# INSTRUCTPROTEIN: ALIGNING HUMAN AND PROTEIN LANGUAGE VIA KNOWLEDGE INSTRUCTION

## ABSTRACT

Large Language Models (LLMs) have revolutionized the field of natural language processing, but they fall short in comprehending biological sequences such as proteins. To address this challenge, we propose InstructProtein, an innovative LLM that possesses bidirectional generation capabilities in both human and protein languages: (i) taking a protein sequence as input to predict its textual function description and (ii) using natural language to prompt protein sequence generation. To achieve this, we first pre-train an LLM on both protein and natural language corpora, enabling it to comprehend individual languages. Then supervised instruction tuning is employed to facilitate the alignment of these two distinct languages. Herein, we introduce a knowledge graph-based instruction generation framework to construct a high-quality instruction dataset, addressing annotation imbalance and instruction deficits in existing protein-text corpus. In particular, the instructions inherit the structural relations between proteins and function annotations in knowledge graphs, which empowers our model to engage in the causal modeling of protein functions, akin to the chain-of-thought processes in natural languages. Extensive experiments on bidirectional protein-text generation tasks show that InstructProtein outperforms state-of-the-art LLMs by large margins. Moreover, InstructProtein serves as a pioneering step towards text-based protein function prediction and sequence design, effectively bridging the gap between protkbein and human language understanding.

## 1 INTRODUCTION

The landscape of Natural Language Processing (NLP) research, and indeed the broader Artificial Intelligence (AI) community, has recently been revolutionized by generative Large Language Models (LLMs) (Peters et al., 2018; Devlin et al., 2019; Brown et al., 2020), such as ChatGPT (Ouyang et al., 2022). The expansion of parameter size and training corpora has empowered these models to acquire versatile, general-purpose data representations that seamlessly transcend linguistic tasks encompassing comprehension and generation in a multitude of languages. Beyond natural languages (a.k.a., human languages), recent investigations have illuminated the potential of these LLMs to serve as a versatile interface for processing multimodal data, including but not limited to images, videos and speech (Chen et al., 2021; Reed et al., 2022; Gong et al., 2023; Huang et al., 2023).

However, general LLMs fall short of capturing the intricate realm of biological sequences, a domain abundant with its own unique linguistic nuances. For example, existing LLMs like ChatGPT cannot understand biological sequences when they are asked to predict the family of proteins (see Figure 1). The biological sequences, particularly proteins, represent a distinctive facet of what could be referred to as "life language", exerting a significant influence on signal transduction pathways, enzymatic catalysis, and gene regulation (Lee & Yaffe, 2016; Huber, 2001; Südhof, 1995; Durek & Walther, 2008; Luzarowski et al., 2021; Jiang et al., 2022).

To unlock the potential within LLMs for deciphering proteins, researchers have put rich efforts into developing protein language models (PLMs) (Alley et al., 2019; Elnaggar et al., 2021; Rives et al., 2021; Rao et al., 2021; Lin et al., 2023). These specialized models are tailored to ingest amino acid sequences as inputs, predict protein functionalities, or even design de novo proteins. Notwithstanding, it is crucial to highlight that while PLMs exhibit competence in comprehending amino acid sequences, they are unable to grasp the complexities of human languages. A recent research trend (Abdine

et al., 2023; Luo et al., 2023) has explored models that accept both protein sequences and textual descriptions as input, aiming to enhance the protein function prediction ability. Nevertheless, these endeavors to align the realms of protein and human languages are unidirectional and remain in their nascent stages; they fall short of being able to generate protein sequences based on textual instructions. In essence, there exists an unaddressed void in the current landscape of LLMs, wherein the ability to swiftly traverse between human and protein languages.

To enable an LLM to adeptly comprehend both human and protein languages, we contend that the limitations imposed by existing models primarily stem from their training corpora. Notably, many existing models are trained on either human languages or protein sequences, rendering them proficient in only one of these linguistic realms. This unilateral training approach is insufficient to imbue an LLM with a comprehensive vocabulary encompassing both languages. Moreover, it is important to recognize that the existing protein-text corpus used in previous studies (Luo et al., 2023; Abdine et al., 2023; Xu et al., 2023; Taylor et al., 2022) has its limitations. (1) The imbalance of annotations: Researchers tend to focus on well-studied proteins, leading to a significant disparity in the availability of annotations (Kustatscher et al., 2022). Training LLMs directly on such a corpus introduces model bias, which ultimately results in suboptimal performance. (2) The absence of instructional signals: Protein-related textual content is primarily comprised of descriptive narratives, often devoid of instructional signals specifically designed for training LLMs. This inherent disparity obstructs a holistic understanding of a wide range of tasks, ultimately resulting in subpar zero-shot performance (Wei et al., 2022a). **In short, the fundamental hurdle of current LLMs involves curating an elaborate training corpus that seamlessly bridges the gap between human and protein languages.**

> **Protein Design**
>
> **Instruction:** I would like a protein that has bifunctional inhibitor/plant lipid transfer protein/seed storage helical domain.
>
> **Output (*ChatGPT*):** Creating a custom protein with specific functional domains would typically involve genetic engineering and molecular biology techniques, and it's a complex and specialized process that requires a laboratory and expertise in the field.
>
> **Output (*InstructProtein*):** One of the protein that meets the demand is MASVKSSSSSSSSSFISLLLLILLVIVLQSQVIE...
>
> ---
>
> **Protein Understanding**
>
> **Instruction:** MFTGGGTIALIERLATSWLTAIRLILSWHPIHAPNRNQ... What family is the protein in?
>
> **Output (*ChatGPT*):** The provided protein sequence is a string of amino acids, and based on the sequence alone, it is not possible to definitively determine the family or function of the protein.
>
> **Output (*InstructProtein*):** The protein is in marek disease virus, lorf3 family.

Figure 1: An example of bidirectional generation by LLMs between human and protein languages. ChatGPT fails to provide an accurate response while the proposed InstructProtein offers a reasonable solution.

In this work, we introduce **InstructProtein**, a pioneering study that aligns human and protein languages through knowledge instruction, leading to the first LLM with bidirectional generation capabilities between these two languages. Specifically, to equip LLMs with the ability to understand protein language, InstructProtein adopts a two-step training approach. It initiates with pre-training on protein and natural language corpora, followed by finetuning with the established protein knowledge instruction dataset. To construct such an instruction dataset, we first transform raw protein-text corpora into a structured knowledge graph (KG). Inspired by the idea of chain-of-thoughts, we enrich KG with knowledge causal modeling, which involves establishing causal relationships between triples, indicating causality within annotations. We then propose a debiased sampling strategy to select KG triples, effectively addressing the issue of annotation imbalance. Finally, we mimic KG completion tasks, leverage general LLMs to convert KG triples into instructions, and conduct supervised instruction tuning. Extensive experiments have demonstrated that the introduced protein knowledge instructions significantly improve the performance of LLMs on protein understanding and design tasks. Our contributions can be summarized as follows:

1. We propose InstructProtein, an innovative LLM that enables bidirectional generation between protein and human languages, effectively filling the gap between the two languages.

2. We introduce a protein instruction generation framework with knowledge graphs, resulting in the first high-quality protein instruction dataset for tuning LLMs.

3. The InstructProtein outperforms state-of-the-art LLMs by a substantial margin, serving as a pioneering step toward text-guided protein function prediction and sequence design.

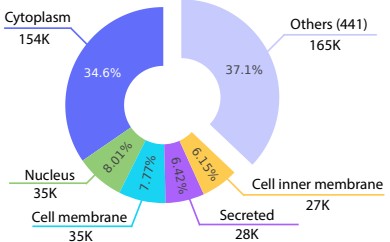

| **Models** | **Prediction** | | | |
|---|---|---|---|---|
| | Cytoplasm | Nucleus | Cell membrane | Others |
| OPT | 2 | 115 | 1691 | 0 |
| LLaMA | 0 | 1806 | 2 | 0 |
| Galactica | 1807 | 1 | 0 | 0 |
| Alpaca | 1808 | 0 | 0 | 0 |

Figure 2: We visualized the top-5 subcellular location categories and their respective proportions, in comparison to the least frequently used annotations, which accounted for only 0.000224%.

Table 1: The results of querying existing LLMs for factual knowledge. We prompt LLMs to predict subcellular location, but their results are biased to a certain category, which suggests that these LLMs have been contaminated by annotation imbalance.

## 2 A CLOSER LOOK AT ANNOTATION IMBALANCE

Much of life science research is dedicated to unraveling the biological functions of proteins. While certain proteins, such as the well-studied tumor suppressor p53 (Dolgin, 2017), have undergone extensive investigation, there still exist tens of thousands of proteins remain categorized as understudied. This phenomenon implies an imbalance in protein function annotation. To clearly illustrate this problem, we take the subcellular location as an example, and show its annotation distribution in Figure 2. The results reveal a notable concentration of research attention on proteins residing in the cytoplasm, while other subcellular locations lack comprehensive labeling and study.

The annotation imbalance has a detrimental effect on the performance of existing LLMs. To demonstrate this, we collect the same number of proteins in each subcellular location category from UniProtKB (Consortium, 2019), resulting in 1,808 proteins in total, and prompt LLMs to predict the subcellular location. The outcomes of LLMs are presented in Table 1, from which one can observe that these LLMs are biased in a certain category, due to the annotation imbalance in the training corpus of LLMs.

## 3 INSTRUCTPROTEIN

This section presents the methodological details of InstructProtein. We first pre-train it in a self-supervised manner on natural language corpus and protein sequence datasets respectively, and then conduct supervised tuning using the created knowledge instruction dataset.

### 3.1 MULTILINGUAL PRE-TRAINING

InstructProtein is designed to comprehend both the protein and human languages. An intuitive approach involves incrementally pre-training an LLM using the protein corpus $\mathcal{P}$ and text sequences $\mathcal{T}$. Given an unsupervised corpus of tokens $\mathcal{X} = \{x_1, x_2, \ldots, x_n\} \in \mathcal{P} \cup \mathcal{T}$, the training objective of a generative LLM (e.g., OPT (Zhang et al., 2022a)) is defined as

$$L(\mathcal{X}) = \sum_i \log P(x_i | x_{i-k}, \ldots, x_{i-1}; \theta), \tag{1}$$

where the prediction of each token depends on previous tokens $x_{<i}$, $k$ is the context window size, and the conditional probability $P$ is modeled using a neural network parameterized by $\theta$.

### 3.2 INSTRUCTION TUNING

After pre-training, the model acquires an extensive comprehension of both natural language and protein sequences; however, it still falls short in achieving alignment between these two different languages. We fill this gap through supervised instruction tuning.

#### 3.2.1 KNOWLEDGE INSTRUCTION GENERATION

We propose an instruction generation method based on knowledge graphs (KGs) and LLMs, aiming to construct a factual, logical, diverse, and well-balanced protein instruction dataset. Figure 3 illustrates the pipeline of three instruction generation frameworks. Conventional approaches directly

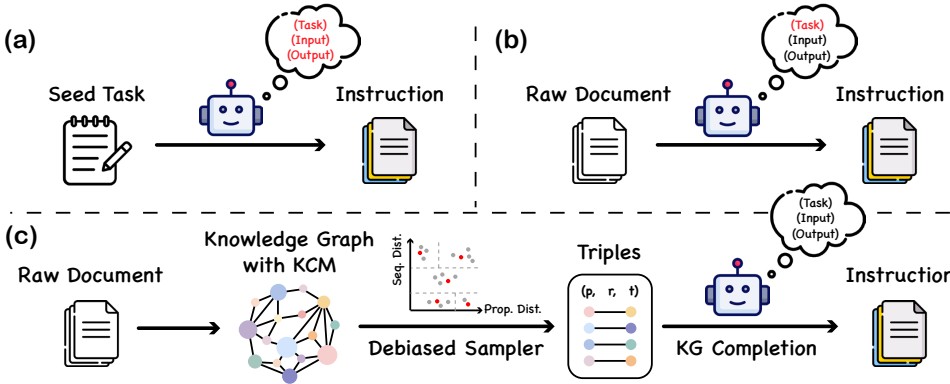

Figure 3: Overview of Instruction generation methods. The red text represents the fields that rely on internal knowledge of LLMs. (a) Given a set of seed tasks, prompting an LLM to produce new instruction data.(b) Utilizing LLMs to generate the instruction data corresponding to the contents in raw documents. (c) The proposed knowledge graph (KG)-based instruction generation framework. We first construct a KG with knowledge causal modeling (KCM), and introduce a debiased sampler to pick the informative triples, which are then translated into instruction data through the use of LLMs in conjunction with KG completion tasks.

utilize LLMs to generate instruction data from seed tasks or raw documents, which may introduce hallucination and bias. In the proposed method, KGs are incorporated as intermediaries to address these limitations. In specific, a KG encompassed with knowledge causal modeling is constructed to provide factual protein knowledge, based on which a debiased sampling strategy is proposed to pick KG triples. An LLM (e.g., ChatGPT) then translates the samples into instruction data and enriches them with a wide range of expressions.

**KG Construction.** We use UniProtKB as our data source to construct the protein knowledge graph denoted as $\mathcal{G} = \{\mathcal{P}, \mathcal{R}, \mathcal{T}\}$. Here, $\mathcal{P}$, $\mathcal{R}$, and $\mathcal{T}$ are sets of protein sequences, relations, and textual annotations. Note that the textual description of proteins in UniProtKB is structured, making it easy to transform them into a knowledge graph. In our pursuit of enhancing the quality of the instruction dataset, we augment KG to provide informative relationships. Borrowing ideas from chain-of-thoughts (Wei et al., 2022b), we recognize that a logical chain also exists within protein annotations. For example, the biological processes in which a protein can participate are intricately linked to its molecular function and subcellular location, with the molecular function itself being influenced by the protein's domain. To represent this causal chain of protein knowledge, we introduce a novel concept called Knowledge Causal Modeling (KCM). Specifically, a knowledge causal model comprises multiple interconnected triples organized in a directed acyclic graph, where the edge direction signifies causal relationships. This graph organizes the triples, moving from the micro-level, encompassing characteristics of protein sequences (e.g., domains), to the macro-level, encompassing biological functions. In Figure 4, we show an example of KCM retrieved from InterPro (Paysan-Lafosse et al., 2023) based on a given triple.

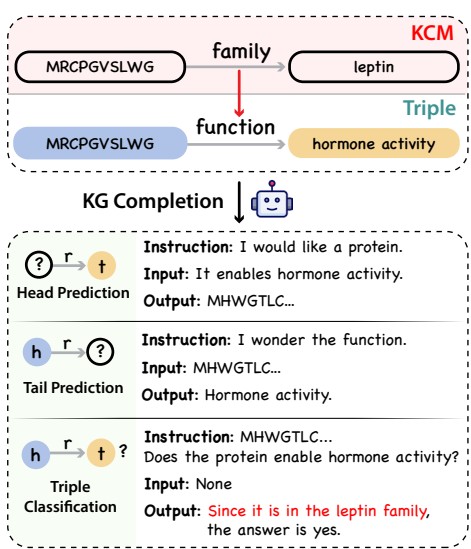

Figure 4: An example of converting a KG triple to instructions. Given a triple with KCM, we use an LLM cooperated with KG completion tasks to generate factual, logical, and diverse instructions.

**KG Triple Sampling.** To generate instruction data, we need to sample triples from the constructed KG. Considering the annotation imbalance problem in the KG, we propose a debiased sampling strategy as an alternative to uniform sampling. In specific, we first group proteins together based on their sequence and property similarities, and then uniformly pick triples within each cluster.

To access sequence similarity, we employ MMseqs2 (Steinegger & Söding, 2017) to calculate the editing distance $d_{\text{seq}}(\cdot, \cdot)$ (see Appendix A.2.2). For property similarity, since the protein properties are extensive and many of them remain unexplored, we only consider the known annotations in KG when computing the property similarity. Specifically, given an annotation $t$ and a relation $r$, we denote $C_t = \{p : p \in \mathcal{P} \wedge (p, r, t) \in \mathcal{G}\}$ and $C_{/t} = \{p : p \in \mathcal{P} \wedge (p, r, t) \notin \mathcal{G}\}$ are the protein set based on the presence or absence of $t$. The basic idea is to maximize agreement within $C_t$ and minimize agreement between $C_t$ and $C_{/t}$, via optimizing protein KG embeddings. In practice, we minimize a margin-based ranking criterion over the knowledge graph:

$$\mathcal{L} = - \sum_{p_t \in C_t, p_{/t} \in C_{/t}} [\log \sigma(\gamma - d_{\text{prop}}(\boldsymbol{p}_t, \boldsymbol{t} + \boldsymbol{r})) + \log \sigma(d_{\text{prop}}(\boldsymbol{p}_{/t}, \boldsymbol{t} + \boldsymbol{r}) - \gamma)], \qquad (2)$$

where $\boldsymbol{p}, \boldsymbol{r}, \boldsymbol{t} \in \mathbb{R}^k$ ($k$ is a hyperparameter) are embeddings of proteins, relations, and annotations, $\sigma$ is the sigmoid function, and $\gamma$ is the margin. $d_{\text{prop}}(\cdot, \cdot)$ is a dissimilarity measure of properties, which is implemented as the $\ell_1$-norm.

We define the threshold of sequence and property similarities as $\delta_{\text{prop}}$ and $\delta_{\text{seq}}$, respectively. We denote two proteins to be similar $p_1 \simeq p_2$ as $d_{\text{seq}}(p_1, p_2) < \delta_{\text{seq}}$ and $d_{\text{prop}}(\boldsymbol{p}_1, \boldsymbol{p}_2) < \delta_{\text{prop}}$. $\mathcal{C} = \{C_1, \ldots, C_m\}$ represents the aggregation of proteins with $m$ clusters, and the cluster $C_i$ can be formulated as:

$$C_i = \{p : \exists p' \in C_i, p \simeq p' \wedge \forall \rho \in C_{j \neq i}, p \not\simeq \rho\}. \qquad (3)$$

Then, the probability of sampling a triple $(p, r, t)$ is:

$$P((p, r, t)) = \frac{1}{m} \times \frac{1}{||C_i||} \times \frac{1}{||p||}, \qquad (4)$$

where $p \in C_i$, $||C_i||$ denotes the size of $C_i$, and $||p||$ are the number of annotations on $p$.

**KG Triple to Instruction.** By employing the debiased sampling strategy, one can sample a large number of well-balanced KG triples. We then focus on translating these triples into instruction data. While the generation of creative tasks requires domain knowledge, the KG completion tasks offer a comprehensive template for proposing domain-specific tasks based on triples. Therefore, we simulate KG completion, and employ general LLMs (e.g., ChatGPT) to transform KG triples with retrieved KCM into instruction data, which contains three fields: an instruction describing the task, an input argument that instantiates the instruction, and an output result reflecting a correct execution of the instruction given the input arguments. Figure 4 shows an example of converting the triple to instructions. The detailed implementation is depicted in Appendix 8.

### 3.2.2 TUNING LLMs WITH INSTRUCTIONS.

Instruction tuning involves further training LLMs in a supervised manner on an instruction dataset comprising of **(instruction, input, output)**, bridging the gap between the LLMs' next-word prediction objective and users' goal of ensuring adherence to human instructions. With the proposed knowledge instruction dataset $\mathcal{I}$, we finetune the pre-trained LLM to align the protein and human languages. Given an instruction $Z \in \mathcal{I}$ and its tokens $\mathcal{X} = \{x_1, x_2, \ldots, x_n\} \in Z$, the training objective is the same as that defined in Eq.(1).

## 4 EXPERIMENTS

In this section, we evaluate the performance of LLMs in terms of protein sequence understanding and design. To effectively evaluate these two capabilities, we have modified the existing downstream task datasets to facilitate the evaluation of LLMs.

### 4.1 EXPERIMENTAL SETUP

The pre-training corpus contains protein sequences from UniRef100 (Suzek et al., 2015) and sentences from PubMed abstracts. Following the methodology described in Section 3.2.1, we generated an instruction dataset comprising 2.8 million data. Specifically, the protein knowledge graph was constructed utilizing the annotations provided by UniProt/Swiss-Prot(Consortium, 2019), which contains the superfamily, family, domain, conserved site, active site, binding site, location, function,

Table 2: Zero-shot performance on protein sequence understanding.

| Models | Params. | Location | | GO-BP | | GO-MF | | GO-CC | | MIB |
|--------|---------|------|------|------|------|------|------|------|------|------|
| | | Bin | Sub | ACC | AUPR | ACC | AUPR | ACC | AUPR | |
| OPT | 1.3B | 57.52 | 29.06 | 51.83 | 64.76 | 56.10 | 74.50 | 51.94 | 71.90 | 49.40 |
| LLaMA | 7.0B | 57.52 | 29.14 | 56.96 | 61.85 | 54.58 | 58.06 | 51.57 | 53.53 | 50.00 |
| Alpaca | 7.0B | 57.52 | 18.32 | 61.69 | 65.13 | 59.37 | 73.02 | 57.98 | 61.71 | 50.38 |
| Galactica | 1.3B | 57.52 | 18.32 | 55.11 | 57.08 | 61.30 | 61.93 | 51.17 | 54.54 | 51.58 |
| BioMedGPT | 10B | 59.51 | 56.39 | 50.31 | 50.82 | 51.02 | 50.81 | 49.41 | 49.39 | 54.42 |
| InstructProtein | 1.3B | **85.19** | **70.79** | **71.49** | **83.16** | **85.83** | **93.68** | **79.79** | **86.37** | **62.68** |

and involved biological process of proteins. Knowledge causal modeling is sourced from the InterPro (Paysan-Lafosse et al., 2023) and Gene Ontology (Aleksander et al., 2023) database. Note that proteins appearing in downstream tasks have been excluded from the training data. We leverage ChatGPT (Ouyang et al., 2022) to convert triples into instruction data. Detailed experimental setups can be found in Appendix A.3.

## 4.2 PROTEIN SEQUENCE UNDERSTANDING

**Datasets and Metrics.** We evaluate LLMs on three widely-used protein function classification tasks: (1) Protein Localization Prediction, which involves the prediction of the subcellular location of a given protein. We address two subproblems from DeepLoc (Almagro Armenteros et al., 2017), the subcellular localization prediction (Abbr., Sub) with 10 location categories and the binary localization prediction (Abbr., Bin) with 2 location categories; (2) Protein Function Annotation, aiming to predict the correct annotations of proteins. We choose Gene Ontology (GO) dataset Gligorijević et al. (2021), which has three branches: molecular function (MF), biological process (BP), and cellular component (CC). We use the dataset splits under 95% sequence identity cutoff. (3) Metal Ion Binding (MIB) Prediction, a binary classification task where the model needs to determine whether there are metal ion-binding sites in the protein. We use the dataset from Hu et al. (2022).

Similar to reading comprehension problems in NLP, we transform all items in the above datasets into a Question&Answer (QA) format where each item consists of a protein sequence, a question about that protein, and a list of possible answers. LLMs are required to predict which answers are true. Following Brown et al. (2020), we use a classification approach where, for example, only two outputs ("*yes*" and "*no*") are considered and the higher probability one is taken as the model's prediction. **All evaluations are carried out in a zero-shot setting, without few-shot demonstrations.**

**Baselines.** We adopt five state-of-the-art open-sourced LLMs as the baselines. OPT (Zhang et al., 2022a) and LLaMA (Touvron et al., 2023) are trained on massive text corpus, and Alpaca (Taori et al., 2023) is a language model based on instruction-tuned LLaMA. Galactica (Taylor et al., 2022) and BioMedGPT (Luo et al., 2023) are domain-specific LLMs, which are trained on a large curated corpus of humanity's scientific knowledge, such as research papers about proteins and genes. For a fair comparison, we designed a template for each model through prompt engineering so that the model could follow our instructions and output the answers.

**Results.** We present the evaluation results in Table 2. Compared with all baselines, InstructProtein achieves new state-of-the-art performance on all tasks. There are two key observations. First, InstructProtein clearly outperforms the LLMs (i.e., OPT, LLaMA, Alpaca) which are stemmed from natural language training corpora. These results demonstrate that training with the corpus where proteins and natural language coexist is beneficial to LLMs, enhancing their proficiency in protein language understanding. Second, InstructProtein performs consistently better than Galactica and BioMedGPT, despite all of them leveraging UniProtKB as a corpus for natural language alignment with proteins. The results verify that our high-quality instruction data can boost zero-shot performance. It is worth noting that in the protein subcellular localization (bin) task, there exists a severe bias in LLMs (OPT, LLaMA, Alpaca, and Galactica), leading to the classification of all proteins into a single group and resulting in the same accuracy of 57.52%.

## 4.3 PROTEIN SEQUENCE DESIGN

Generating proteins following human instructions is a highly exciting area of research. With the incorporation of protein sequences as part of the language capabilities in LLMs, InstructProtein is

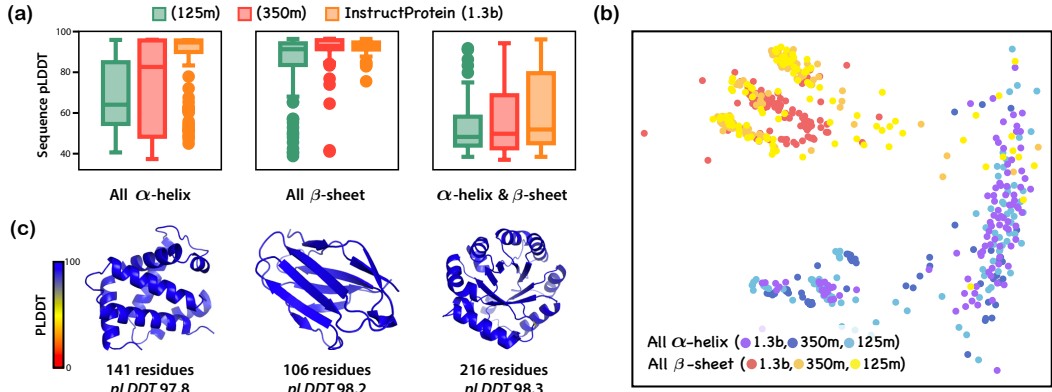

Figure 5: Visualization of structure instruction-based protein sequence de novo design. We prompt our models with different scales (125m, 350m and 1.3b) to generate three kinds of proteins (all $\alpha$-helix, all $\beta$-sheet, and a combination of $\alpha$-helix and $\beta$-sheet), respectively. (a) We visualize the pLDDT of generated sequences predicted by AlphaFold2 to assess the protein foldability. (b) The embeddings of sequences prompted with all $\alpha$-helix and all $\beta$-sheet instructions, which are extracted from ESM2 and visualized by the MDS algorithm. (c) The structure of generated proteins with the highest confidence in each class.

capable of generating protein sequences. However, the lack of standardized computational metrics to assess the quality of generated proteins poses challenges for advancing protein generation models. In this study, we present our endeavor to build a computational evaluation framework.

### 4.3.1 ZERO-SHOT INSTRUCTION-PROTEIN PAIRING

**Datasets and Metrics.** We design an instruction-protein pairing task to assess the consistency between the instruction and the generated protein. Specifically, we employ the dataset proposed by Hou et al. (2018) to provide fold-related instructions and proteins. Given a protein $p$ and the corresponding instruction $Z_0$, we randomly sample other $n$ instructions $\{Z_1, Z_2, ..., Z_n\}$ ($n = 9$ in this experiment), and the likelihood $\mathcal{L}$ of the protein given the various instructions is computed. The minimization of $\mathcal{L}(p|Z_i)$ at $i = 0$ signifies a correct pairing, and vice versa.

**Results.** Table 3 reports the accuracy of the instruction-protein pairing task. One can observe that InstructProtein surpasses the baselines by a large margin. BioMedGPT focuses solely on converting proteins to texts and lacks protein design capabilities. Galactica exhibits limited zero-shot performance in aligning instructions with proteins, since it is trained with narrative protein corpus. These results confirm the superiority of our model in instruction-following for protein generation.

Table 3: Accuracy of instruction-protein pairing.

| Models | Fold Rank | | |
|---|---|---|---|
| | Fold | SuperFamily | Family |
| OPT | 7.79 | 6.45 | 6.68 |
| LLaMA | 9.33 | 5.90 | 10.30 |
| Alpaca | 5.43 | 3.90 | 4.71 |
| Galactica | 11.00 | 10.12 | 10.37 |
| BioMedGPT | - | - | - |
| InstructProtein | **55.57** | **65.07** | **79.24** |

### 4.3.2 PROTEIN SEQUENCE DE NOVO DESIGN

**Designing proteins with specified structures.** We investigate whether InstructProtein could generate new protein sequences that are individually valid and consistent with instructions. SCOPe (Chandonia et al., 2022) classifies protein structures according to the content and organization of secondary structures, including all $\alpha$-helix, all $\beta$-sheet, and the combination of $\alpha$-helix and $\beta$-sheet. We sample 100 sequences from each class and assess the foldability of individual sequences by predicting their corresponding structures using ColabFold (Mirdita et al., 2022; Jumper et al., 2021) and computing the average predicted local distance difference test (pLDDT) across the whole structure (Figure 5 (a)). pLDDT increases with model scale, suggesting that scaling up the parameter size leads to generating more foldable sequences. We leverage ESM2 (Lin et al., 2023) as a feature extractor to obtain the generated all $\alpha$-helix and all $\beta$-sheet protein representations, which are then

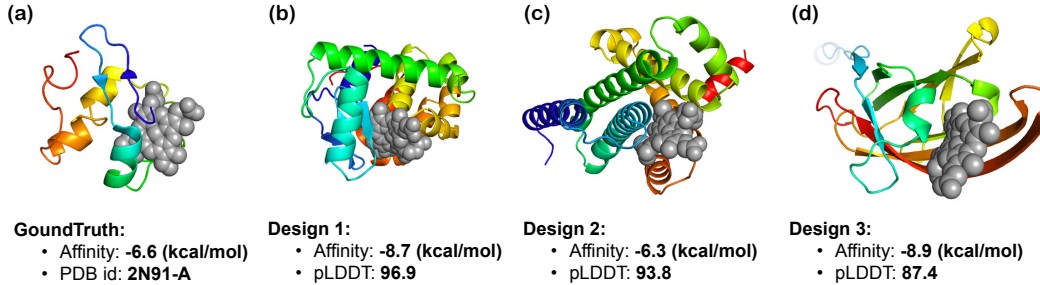

Figure 6: Visualization of functional instruction-based protein sequence de novo design. We prompt our model with the instruction "I would like a protein that enables heme binding". (a) is the ground-truth protein that binds with heme. (b), (c) and (d) are generated proteins with decent binding affinity.

Table 4: Ablation of the proposed sampling strategy and KCM used in knowledge instructions.

| Sampling Strategy | KCM | Location (Sub) | GO (MF) | Fold Rank (Fold) |
|---|---|---|---|---|
| Unclustering | No | 58.12 | 85.58 | 51.98 |
| Seq. Clustering | No | 62.77 | 83.70 | 54.41 |
| Seq.&Prop. Clustering | No | 69.95 | **85.92** | 53.81 |
| Seq.&Prop. Clustering | Yes | **70.79** | 85.83 | **55.57** |

visualized using multi-dimensional scaling (MDS) algorithm (Kruskal, 1964) (Figure 5 (b)). We observe that the representations are divided into two groups according to instructions, indicating the instruction-following ability of the proposed model. We visualize the predicted structure of the proteins with the highest confidence in each class (Figure 5 (c)). These results demonstrate that InstructProtein establishes a close correlation between natural language and protein language, verifying the effectiveness of protein de novo design based on structure-related instruction.

**Designing proteins binding with specified ligands.** To verify the ability to follow function-related instructions, we employ InstructProtein to design heme binders, which are proteins capable of binding to a specific compound, and visualize 3D structures of three generated proteins. In Figure 6, we present the docking result (docked by DiffDock (Corso et al., 2023)), the binding affinity (predicted by Smina (Koes et al., 2013; Trott & Olson, 2010), the lower the better), and the pLDDT score (predicted by ColabFold; the higher the absolute value, the better). We can observe the resulting proteins exhibit notable binding affinity, confirming the efficacy of InstructProtein in heme binder design. We provide more case studies in Appendix A.5.

### 4.4 ABLATION STUDY

In this subsection, we conduct ablation studies on the sampling strategy and knowledge causal modeling (KCM) used in our knowledge instruction generation method. From the results in Table 4, we observe that clustering similar proteins in annotation imbalance-related tasks (Location and Fold Rank) can effectively improve model performance. However, for the GO task without the annotation imbalance problem, the clustering method based on sequence similarity alone makes the model performance decrease. This phenomenon arises due to the occurrence of critical mutations at key sites, leading to significant functional alterations in proteins. Consequently, their resemblance to extensively studied sequences diminishes the likelihood of selection. This leaves the model lacking the ability to distinguish the functionally important sites. Such problems can be avoided by segment clusters based on protein properties. We also observe that the causal relationship between annotations introduced by KCM improves the performance.

## 5 RELATED WORKS

**Large Language Models** (LLMs) have achieved breakthrough performance in NLP (Brown et al., 2020; Rae et al., 2021; Hoffmann et al., 2022; Black et al., 2022; Zhang et al., 2022a; Chowdhery et al., 2022; Touvron et al., 2023). These models, trained via self-supervision on extensive, general

corpora, exhibit proficiency in a multitude of tasks (Hendrycks et al., 2020; Jin et al., 2021; Pal et al., 2022). However, these LLMs are primarily tailored for human language comprehension, limiting their utility in decoding the intricate language of proteins. To bridge this gap, Protein Language Models (PLMs) have garnered significant attention (Alley et al., 2019; Elnaggar et al., 2021; Rives et al., 2021; Rao et al., 2021; Lin et al., 2023; Rao et al., 2020; Meier et al., 2021; Ferruz et al., 2022; Notin et al., 2022). Nonetheless, PLMs confront limitations stemming from their training corpora, lacking factual knowledge of human language. To align protein with human language, multimodal approaches (Abdine et al., 2023; Luo et al., 2023) integrate protein encoders into LLMs within an encoder-decoder framework. Notwithstanding, these architectures predominantly exhibit a unidirectional cross-modal capability, focusing solely on converting protein language to texts. Taylor et al. (2022) treats protein language and human language as a unified modality. However, the use of existing protein-text corpus hinders the alignment of protein and human languages. The proposed InstructProtein represents a pioneering effort with the ability to generate in both human and protein languages, marking a significant advancement in the field of protein understanding and design.

**Instruction Tuning** is a supervised approach to align language models with user intention (Weller et al., 2020; Mishra et al., 2022; Wang et al., 2022; Wei et al., 2021; Sanh et al., 2021; Ouyang et al., 2022). It is worth noting that acquiring large-scale instruction data can be a resource-intensive and time-consuming endeavor, thereby motivating the exploration of automatic data generation techniques. A prevalent strategy (Anaby-Tavor et al., 2020; Andreas, 2020; Kaushik et al., 2019) involves augmenting existing datasets. Alternatively, several fully automatic datasets have been proposed to eliminate the need for labeled data. Schick & Schütze (2021); Ye et al. (2022) advocate for leveraging pre-trained language models to generate comprehensive labeled datasets from scratch, tailored to predefined tasks. Honovich et al. (2023a),Wang et al. (2023) and Honovich et al. (2023b) used pre-trained LLMs to automatically construct instructions by a handful of examples. Fang et al. (2023) and Li et al. (2023) leverage content-first apporaches, in which LLMs construct instruction data by generating raw document-corresponding tasks. However, these methodologies may introduce hallucination and bias into the instruction data. To overcome these limitations, our work incorporates knowledge graphs as intermediaries, resulting in a protein instruction dataset that is factual, logical, diverse, and well-balanced.

**Knowledge Graph** (KG) is often employed to enhance the capabilities of LLMs. A related subfield to our work involves integrating KGs into the input of LLMs. Researchers such as Sun et al. (2021); Liu et al. (2020); Sun et al. (2020); Zhang et al. (2022b) have pursued this avenue by concatenating a KG triplet with corresponding sentences, leveraging language modeling to amalgamate knowledge with textual representations. Our approach also involves utilizing KGs as input, however, a significant difference lies in how LLMs interact with KGs. We focus on generating instruction data using KGs, allowing LLMs to capture insights from instructions rather than relying solely on KG triplets. Moreover, following previous studies (Zhao et al., 2021; Zhang et al., 2023; Ma et al., 2023), we propose a novel KG-based sample strategy to avoid the understudy problem in knowledge-intensive domains.

# 6 DISCUSSION AND CONCLUSION

InstructProtein explores the feasibility of bidirectional generation between human and protein languages within a single large language model. Our approach involved the transformation of a raw protein-text corpus into a structured knowledge graph, from which KG triples were sampled and converted into instructions. This KG-based instruction generation method resulted in a high-quality instruction dataset, facilitating the LLM to align protein language with human language.

Nevertheless, it's important to acknowledge that there are some limitations inherent in our model. One such limitation, shared with large language models, is that InstructProtein encounters challenges with handling numerical values. This limitation hinders our ability to quantitatively characterize proteins, including tasks like thermostability prediction. Besides, the design of a satisfactory protein necessitates meeting a multitude of requirements, such as solubility, stability, and 3D structure. However, our current model is primarily tailored to support protein design based on qualitative descriptions, such as designing proteins within specific protein families. This limitation arises from our instructions exclusively offering qualitative protein descriptions encompassing aspects like family and function, while lacking quantitative annotations concerning elements such as 3D structures, which hold significance in protein design.

In the future, we will incorporate a broader spectrum of instructions, including quantitative descriptions. This extension will empower our model to provide quantitative outputs. These developments will open up new avenues for further advancing the integration of protein and human languages, as well as expanding its practical utility in diverse applications.

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

# A  APPENDIX

## A.1  DETAILED PROTEIN UNDERSTUDYING PROBLEM ANALYSIS

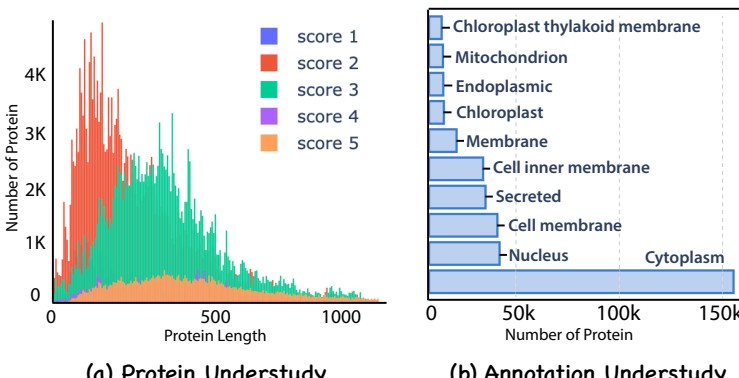

(a) Protein Understudy          (b) Annotation Understudy

Figure 7: The overview of the problem of understudied proteins. (a) We visualized the protein length distribution for different annotation scores. The annotation score provides a heuristic measure of the annotation content (Score 5 is associated with the best-annotated entries, and a score 1 denotes an entry with rather basic annotation.). (b) We visualized the ten most used categories in subcellular location annotations.

Much of life science research is dedicated to unraveling the biological functions of proteins. While certain proteins, such as the well-studied tumor suppressor p53 (Dolgin, 2017), have undergone extensive investigation, tens of thousands of proteins remain categorized as understudied. This classification implies that their biological functions are poorly elucidated, and they lack comprehensive annotation of their molecular properties.

In Figure 7, we present an analysis conducted on UniProtKB/Swiss-Prot, a highly reputable and manually curated protein knowledge repository. Figure 7 (a) depicts the relationship between the distribution of proteins and their annotation scores. These results emphasize the substantial variation in protein distribution corresponding to different annotations. This variance implies that the annotation of proteins is biased. To illustrate this problem more clearly, we analyze the subcellular location annotation. Figure 7 (b) illustrates the distribution of such annotations. The data reveals a notable concentration of research attention on proteins residing in the cytoplasm, with other subcellular locations significantly lacking in comprehensive labeling and study.

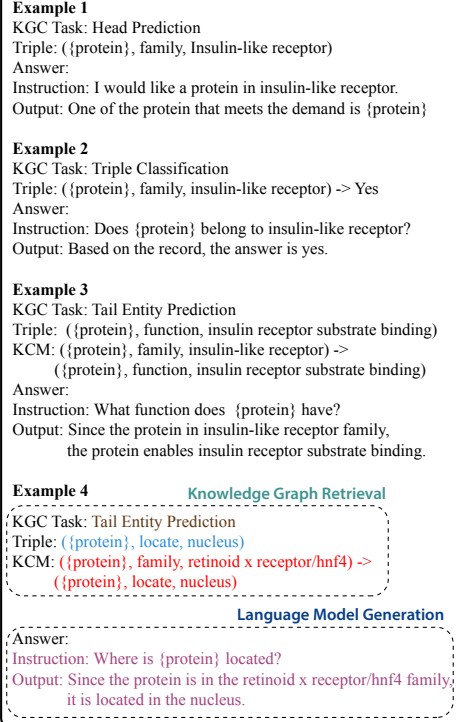

Figure 8: Our data generation prompt. We provide three in-context examples with and external knowledge needed to generate the next instruction data. Purple: One of the model's generations for the given prompt.

## A.2  DETAILED METHOD

### A.2.1  IN-CONTEXT EXAMPLES

Knowledge Instruction relies on examples to teach language models understand how to convert information extracted from the knowledge graph into instruction data. Here we provide our example (Figure 8). We notice that when only two examples of

Table 5: Example of instructions in the Knowledge Instruction dataset. We leverage {protein} as a placeholder for the corresponding protein sequence.

| Task | Instruction |
|---|---|
| Family Generation | Instruction: Can you provide me with a protein belonging to the secretoglobin family? 
 Output: Sure, here's a protein from the secretoglobin family: {protein}. |
| Superfamily Prediction | Instruction: Could you please provide information about the superfamily to which {protein} belongs? 
 Output: {protein} is likely a member of the rho GTPase activation protein superfamily. |
| Domain Generation | Instruction: Could you provide me with a protein that contains the nad(p)-binding rossmann-fold domain? 
 Output: Certainly, here is a protein that contains the nad(p)-binding rossmann-fold domain: {protein}. |
| Family Prediction | Instruction: What family does {protein} belong to? 
 Output: The protein belongs to the alpha-1b-glycoprotein/leukocyteimmunoglobulin-like receptor family. |
| Function Prediction | Instruction: protein Does the protein have the molecular function of zinc ion binding? 
 Output: Yes, based on the information that the protein belongs to the Retinoid X receptor/HNF4 family, it indeed has the molecular function of zinc ion binding. |
| Process Prediction | Instruction: I'm interested in learning about the biological process associated with {protein}. Could you provide information on this? 
 Output: Given its association with the Retinoid X receptor/HNF4 family, {protein} is likely involved in the regulation of DNA-templated transcription. |
| Binding Site Generation | Instruction: Provide me with a protein associated with the IQ motif EF-hand binding site. 
 Output: Sure, the protein that interacts with the IQ motif EF-hand binding site and exhibits a protein binding molecular function is {protein}. |
| Family Prediction | Instruction: Does {protein} belong to nfat activation molecule 1? 
 Output: Based on the record, the answer is yes. |

different expressions are provided for each KGC task, the language capabilities of LLMs are activated, generating a variety of instruction data as illustrated in Table 5

### A.2.2 EDIT DISTANCE ALGORITHM

We denote $A = a_1 a_2 \ldots a_n$ and $B = b_1 b_2 \ldots b_n$ as two sequences to be aligned, where $n$ and $m$ are the lengths of $A$ and $B$, respectively. Before calculating the editing distance, we have to determine the substitution matrix to calculate the replacement score $s(\cdot, \cdot) \in (0, 1]$ and the gap penalty scheme $W_k$, where $k$ is the gap length. Then the distance matrix $H$ can be formulated as:

$$H_{i,j} = \min\{H_{i-1,j-1} + s(a_i, b_j); H_{i-k,j} - W_k; H_{i,j-1} - W_1; 1\} \tag{5}$$

where $H_{k,0} = H0, l = 0$ for $0 \leq k \leq n$ and $0 \leq l \leq m$. We leverage $H_{n,m}/\max(n, m)$ as the sequence distance between $A$ and $B$.

### A.3 DETAILED EXPERIMENTAL SETUPS

We perform incremental training on OPT-1.3b. We wrap the protein sequence with <protein> and </protein> and apply character-based tokenization, treating each amino acid as a single token. For text corpus, we tokenize them using the GPT-2 byte level BPE tokenizer. We utilize Pytorch to conduct experiments with 8 32G V100 GPUs. We use a batch size of $128$ and a context length of $1,024$ tokens. We adopt the Fully Sharded Data Parallel (FSDP) acceleration strategy alongside the fp16 data format. We adopt the AdamW optimizer with $\beta = (0.9, 0.98)$. We set the weight decay to 0.01 and the dropout rate to 0.1. The learning rate increases to 1e-4 for the first 5000 warming-up steps and decays linearly to 0 for the rest of the training steps. We pre-train InstructProtein for the first 40,000 steps, and instruction tune it in the next 20,000 steps.

### A.4 DOWNSTREAM TASK DEFINITION

We list the detailed definition of downstream tasks. {protein} and {label} are used as placeholders. Dataset statistics are summarized in Table 6.

**Subcellular Localization Prediction** is a sequence-level classification task. Each input sequence $x$ is mapped to a label $y$ which represents the subcellular location.

- Prompt template (InstructProtein, OPT, LLaMA, Alpha, BioMedGPT): {protein} Instruction: What cellular components is the protein located in?
- Prompt template (Galactica): {protein} ## Subcellular Location
- Label words (sub): {0: "plasma membrane", 1: "cytoplasm", 2: "endoplasmic reticulum", 3: "golgi", 4: "vacuole", 5: "mitochondrion", 6: "nucleus", 7: "peroxisome", 8: "chloroplast", 9: "extracellular"}
- Label words (bin): {0: ["plasma membrane", "golgi", "vacuole", "endoplasmic reticulum"], 1: ["extracellular", "peroxisome", "nucleus", "cytoplasm", "mitochondrion", "chloroplast"]}

**Protein Function Annotation** is a sequence-level classification task to annotation protein with functional labels. Each example consists of a protein, a label. They system must predict whether the label belongs to the protein.

- Prompt template: {protein} Instruction: Does the protein associate with label?
- Label words: {0: "No", 1: "Yes"}

**Metal Ion Binding Prediction** is a sequence-level classification task to predict whether a protein can bind to ion.

- Prompt template: {protein} Instruction: Does the protein associate with metal ion binding?
- Label words: {0: "No", 1: "Yes"}

**Instruction-Protein Pairing Accuracy** probe the insturction-following capabilities in protein generation. Protein are decoded under 10 different instructions (9 randomly sampled instructions and 1 true corresponding instruction). The system must predict which one is the most relevant instruction.

- Prompt template: Instruction: I would like a protein that is in {label}. Output: One of the protein that meets the demand is {protein}"

Table 6: Dataset Statistics for downstream tasks.

| Dataset | # Test |
| --- | --- |
| Subcellular Localization Prediction - *bin* | 1,749 |
| Subcellular Localization Prediction - *sub* | 2,773 |
| Protein Function Annotation - *Biological Process* | 104,794 |
| Protein Function Annotation - *Molecular Function* | 22,372 |
| Protein Function Annotation - *Cellular Component* | 38,594 |
| Metal Ion Binding Prediction | 1,332 |
| Instruction-Protein Pairing Accuracy - *Fold* | 718 |
| Instruction-Protein Pairing Accuracy - *Family* | 1,272 |
| Instruction-Protein Pairing Accuracy - *Superfamily* | 7,408 |

## A.5 MORE EXAMPLES

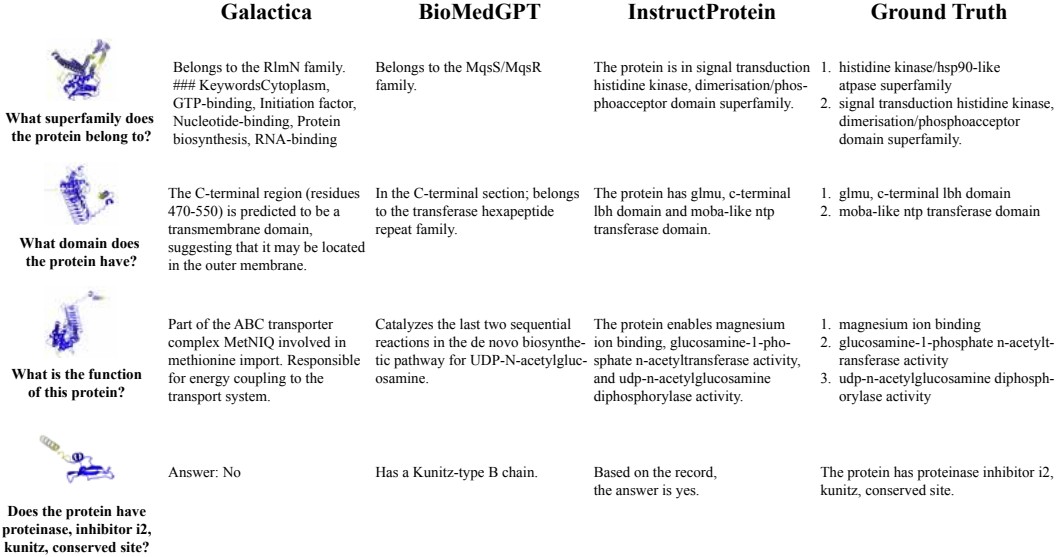

Figure 9: More examples of protein understanding.

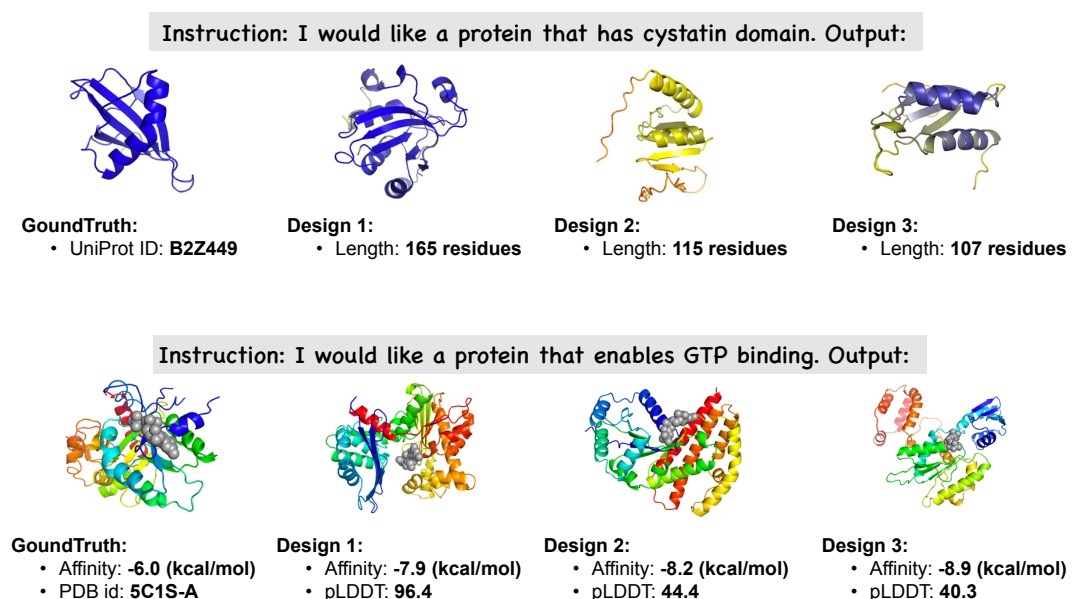

Figure 10: More examples of function-instruction-based protein de novo design.

**Instruction: I would like a protein that is in metallothionein family. Output:**

**GoundTruth:**
• UniProt ID: **A0A024R6T4**

**Design 1:**
• Length: **61 residues**

**Design 2:**
• Length: **61 residues**

**Design 3:**
• Length: **54 residues**

**Instruction: I would like a protein that is in retroviral VpR/VpX protein family. Output:**

**GoundTruth:**
• UniProt ID: **A0A023HIS7**

**Design 1:**
• Length: **126 residues**

**Design 2:**
• Length: **96 residues**

**Design 3:**
• Length: **96 residues**

**Instruction: I would like a protein that is in SsrA–binding protein family. Output:**

**GoundTruth:**
• UniProt ID: **A0ALD2**

**Design 1:**
• Length: **153 residues**

**Design 2:**
• Length: **150 residues**

**Design 3:**
• Length: **151 residues**

**Instruction: Instruction: I would like a protein that is in kappa casein family. Output:**

**GoundTruth:**
• UniProt ID: **P02668**

**Design 1:**
• Length: **98 residues**

**Design 2:**
• Length: **145 residues**

**Design 3:**
• Length: **192 residues**

Figure 11: More examples of family-instruction-based protein de novo design (colored by pLDDT).

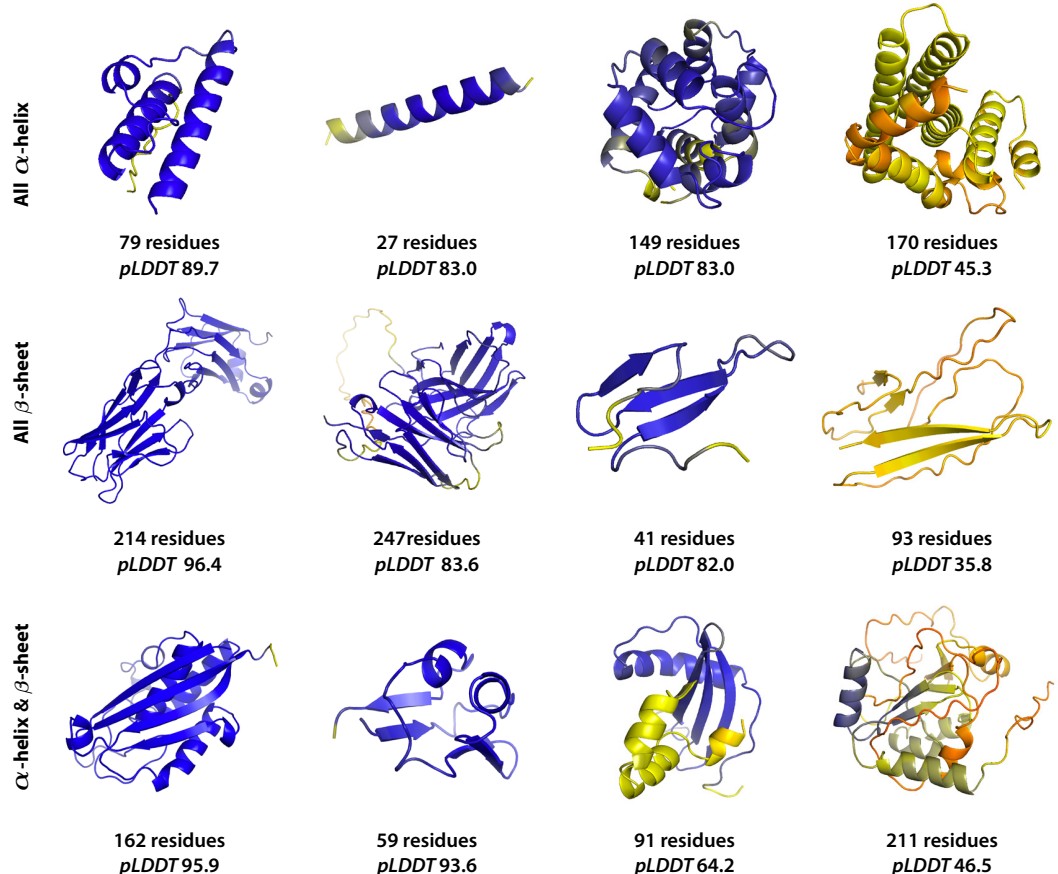

Figure 12: More examples of structure-instruction-based protein de novo design (colored by pLDDT).

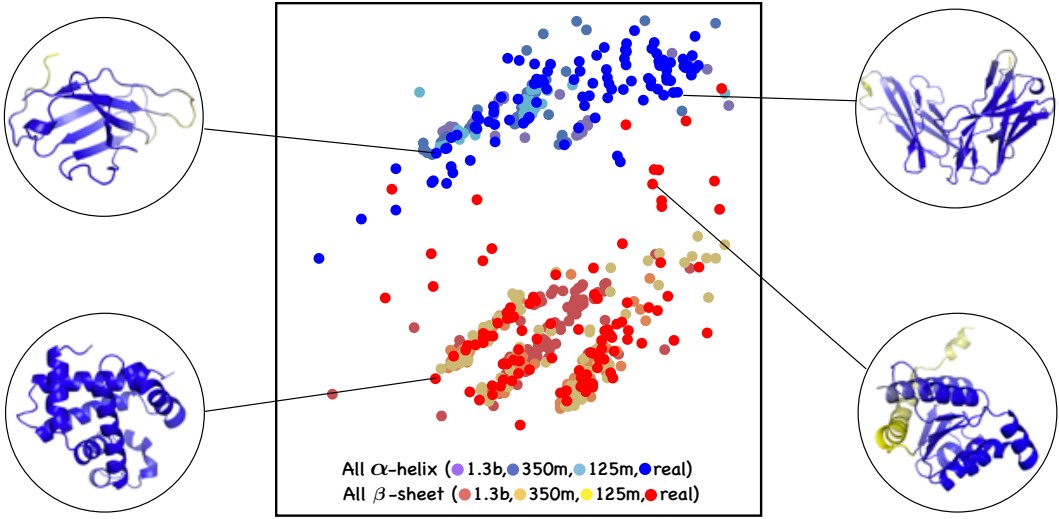

Figure 13: Visualization of the embeddings of proteins designed based on structure instructions.

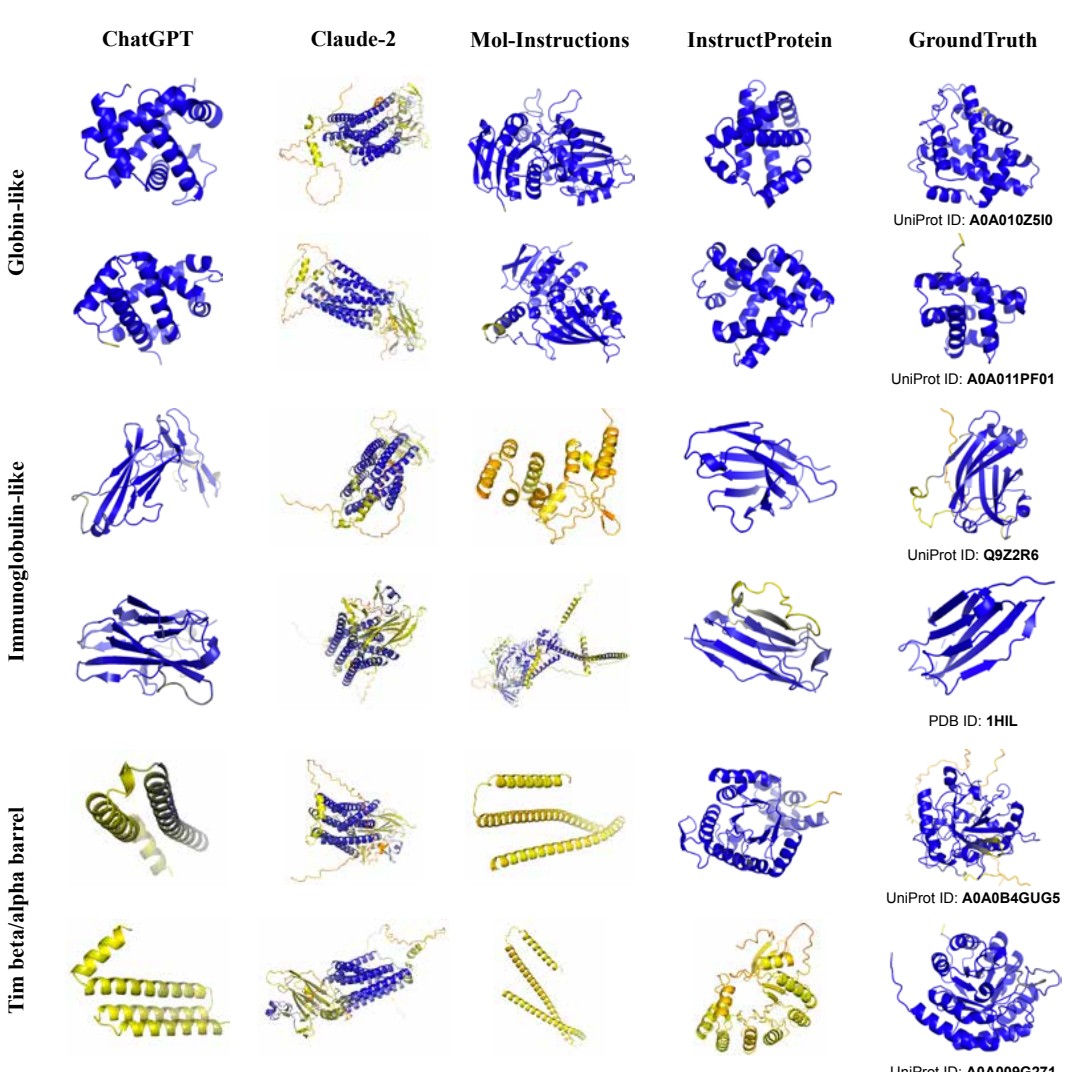

Figure 14: Visualize proteins generated by ChatGPT, Claude-2, InstructProtein.

**Prompt:**
I would like a protein that enables metal ion binding.

**Design 1:**
MNKKLVVFDTTLRDGEQSPGASMTKEEKVRIAKQLERMKVDVIEAGFAAASEGDFEAVKAIAQAIKHSTVCSLARANDK
DIARAAEALKPANSFRIHTFIATSALHMEKKLRMTPDQVYEQAVKAVRFARKFTDNVEFSPEDAGRSDPEFLCQVVEAV
INEGASTINIPDTVGYSIPSQFGQLIATIRGRVPNSDKAIWSVHCHNDLGMAVANSLAGVKIGGARQIECTINGLGERA
GNCSLEEIVMAVKTRKDYFGLDVGIDAQHILPASKLVSQITGFVVQPNKAVVGANAFAHASGIHQDGVLKARDTYEIMR
AEDVGWSANKIVLGKLSGRNAFKQRLQELGVQLESEAELNAAFMRFKDLADRKAEIFDEDIIAIVSEECALAQEQEHFK
FVSLSQRSETGEQPQAKVVFAVEGKEVTGEARGNGPVDATFNAIEGEVGSGSELLLYSVNAISTGTQAQGEVTVRLSKS
GRIVNGVGTDPDIVAASAKAYISALNKLHSKDDKLNPQRA

**Design 2:**
MSDTLFRQLDILRAELDQHNYRYYVDDAPSVPDAEYDRLMQELVALEAEHPELLTPDSPTQRVGGEALAAFRQVRHRIP
MLSLGNAFEDGDLEAFKRRVDRLTGELPGGPLELVCEPKLDGLAVSLLYRDGQLVRGATRGDGTTGEEVSTNVRTIRNI
PLRLHGKGWPAVLEVRGEVYMSRAGFDRLNAAQAEAGGKTFANPRNAAAGSLRQLDSKITASRPLEFCCYGIGQVSECF
DGHQQGLLARLRAWGLPINQELRHAEGLEACLAFYRKVGERRASLPYEIDGAVFKLDERARQRLGFRARAPRWAIAHKF
PAMEELTELLDVEFQVGRTGAITPVARLKPVQVAGVTVSNATLHNMDEIARLGLRPGDRVIVRRAGDVIPQVVQVVLER
RPEGARPIPLPAHCPVCGSEAVREPGEVAVRCVGGLYCEAQRKEALRHFASRRALDVEGLGDKLVDQLVDRELVGSPAD
LYRLTYEQLVGLEGFAEVSSKKLAEAIASSRQPSLARFIYALGIREVGEATAKDLARHFGTLDALMAADEEALLAVPDV
GPVVAHNIRHFFAQPHNLEVVDALLTAGVHWPQVEAVADEAPLAGQTWVLTGTLDSMSRDDAKAQLEALGARVSGSVSK
KTAAVVAGEAAGSKLAKAQELGVKVLDEEELLRLLAAP

**Design 3:**
MERKSKNPELSSKAFAIAYCAVCGLSRSRNQVFIHKKGLNICRQCFYKKFVSEIISEEKIRQKQRTATKTLPLKDCSDD
IKLKIDPKLKQDKTFEVIEDYEEEIKIEDTIWETNAINEISTCSVCGREEGGFFEAVVQVRAADLLYPTGAQENDELNR
QVREKLQSLRQKVATAKKEFNHIDEYAKHIQKILEEKGVSHDLVRLSTYDLAAAQSLGAVPSEGQKKPGTDEKLRIPRS
GLGLKASLISTYADKVRAARAKLKKAGDIDAQFINGIIKILVPMRPSAVHLDLLENLGCKKVLPENPQIDVLSTRKYPT
LLHGGEPLKEAPLIAPSLIQLLKPYMEGQGLVVVDSFLLSSSRVLRLLKEGALDYLIIEGKYRILSIEKGKFEIDFEGK
PDCVGGSEDLAKLAEVVGAEYLGEDALEVVKLGNQDVTKVKNVQSLEIEEIIRNESIKVKTILQEKYEDIESIIKTKER
LDKLKKEKRKKEIKKNKLTDNANKLEPIETEQELQKLESRNIPLEELLELTNELRPLILANVAERVRKELEIQKIPTEK
NE

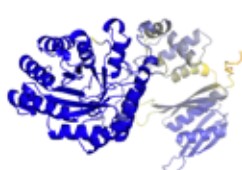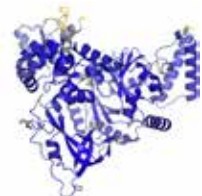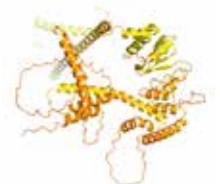

**Design 1:**
  **Similar Protein: A0A1Z5HSR0**
  **Metal Ion Binding: Enable**

**Design 2:**
  **Similar Protein: A0A7C4E6S4**
  **Metal Ion Binding: Enable**

**Design 3:**
  **Similar Protein: None**

Figure 15: Visualize InstructProtein designed proteins that enable metal ion binding.

MHHHHHHSSGVDLGTENLYFQSNAMDFPQQLEACVKQANQALSRFIAPLPFQNTPVVETMQYGALLGGKR
LRPFLVYATGHMFGVSTNTLDAPAAAVECIHAYSLIHDDLPAMDDDDLRRGLPTCHVKFGEANAILAGDALQ
TLAFSILSDANMPEVSDRDRISMISELASASGIAGMCGGQALDLDAEGKHVPLDALERIHRHKTGALIRAAVR
LGALSAGDKGRRALPVLDKYAESIGLAFQVQDDILDVVGDTATLGKRQGADQQLGKSTYPALLGLEQARKK
ARDLIDDARQALKQLAEQSLDTSALEALADYIIQRNK

**User:** Does the protein enable transferring alkyl or aryl groups, other than methyl groups? [Yes/No]
**ChatGPT**: No. The provided amino acid sequence does not contain known motifs or domains associated with enzymes
that transfer alkyl or aryl groups. The sequence lacks characteristics of methyltransferases, which are enzymes involved
in the transfer of methyl groups. Additionally, there are no specific functional annotations or signatures indicating a role
in alkyl or aryl group transfer. Therefore, based on the given sequence, it is not likely that the protein enables the
transfer of alkyl or aryl groups other than methyl groups.
**GroundTruth**: Yes

**User:** Does the protein enable (2E,6E)-farnesyl diphosphate synthas? [Yes/No]
**ChatGPT**: No. The absence of such motifs or homology in the provided sequence suggests that this protein is not likely
to be (2E,6E)-farnesyl diphosphate synthase. It's important to note that a definitive answer would require experimental
validation or more detailed bioinformatic analysis.
**GroundTruth**: Yes

**User:** Does the protein enable poly(A)-specific ribonuclease? [Yes/No]
**ChatGPT**: No, based on the provided amino acid sequence, there is no indication or motif suggesting that the protein
enables poly(A)-specific ribonuclease. The sequence doesn't contain known motifs associated with poly(A)-specific
ribonuclease activity.
**GroundTruth**: No

MAHHHHHHMALVSMRQLLDHAAENSYGLPAFNVNNLEQMRAIMEAADQVNAPVIVQASAGARKYAGAPF
LRHLILAAVEEFPHIPVVMHQDHGASPDVCQRSIQLGFSSVMMDGSLLEDGKTPSSYEYNVNATRTVVNFSH
ACGVSVEGEIGVLGNLETGEAGEEDGVGAAGKLSHDQMLTSVEDAVRFVKDTGVDALAIAVGTSHGAYKFT
RPPTGDVLRIDRIKEIHQALPNTHIVMHGSSSVPQEWLKVINEYGGNIGETYGVPVEEIVEGIKHGVRKVNIDT
DLRLASTGAVRRYLAENPSDFDPRKYLGKTIEAMKQICLDRYLAFGCEGQAGKIKPVSLEKMASRYAKGELN
QIVK

**User**: Does the protein enable fructose-bisphosphate aldolase? [Yes/No]
**ChatGPT**: No. The provided sequence appears to be a longer peptide with no recognizable motifs or domains associated
with aldolase activity. Additionally, the sequence does not align with known fructose-bisphosphate aldolase sequences.
**GroundTruth**: Yes

**User**: Does the protein enable aldehyde-lyases? [Yes/No]
**ChatGPT**: No. The provided amino acid sequence does not exhibit clear motifs or conserved residues associated with
aldehyde-lyases. Aldehyde-lyases are enzymes that catalyze the cleavage of carbon-carbon bonds in aldehydes.
Typically, these enzymes have specific motifs and active site residues that are essential for their catalytic activity.
The sequence you provided does not contain these characteristic features associated with aldehyde-lyases.
**GroundTruth**: Yes

**User**: Does the protein enable non-chaperonin molecular chaperone ATPase? [Yes/No]
**ChatGPT**: No. The provided amino acid sequence does not exhibit characteristics or motifs commonly associated with
non-chaperonin molecular chaperone ATPase activity. Molecular chaperones, including chaperonin ATPases, typically
possess specific domains and motifs involved in ATP binding and hydrolysis, as well as substrate binding. The absence
of such features in the given sequence suggests that it is not likely to function as a non-chaperonin molecular chaperone
ATPase.
**GroundTruth**: No

Figure 16: ChatGPT prediction results of whether a protein can catalyze a reaction.

