# OpenReview forum: "InstructProtein: Aligning Human and Protein Language via Knowledge Instruction"
_ICLR.cc/2024/Conference — ICLR 2024 Conference Withdrawn Submission_

### Official Review · Reviewer_DyDT · 2023-10-31

**Soundness:** 2 fair
**Presentation:** 3 good
**Contribution:** 3 good
**Rating:** 6
**Confidence:** 3

**Summary:**

While showing revolutionary abilties in classical language sequences processing, Large Lanuage Models fall short at processing protein sequences, for example, protein function annotation. This paper makes effort to allow LLMs work better in protein processing tasks.

The authors claim the main challenge lies on the corpus diffenrence between the areas of human and protein languages.
To align the two languages, the authors propose (1) incrementally pre-training a LLM using both protein corpus and text corpus and (2) supervised instruction tuning LLMs by collecting and utilizing a knowledge-graph-enhanced instruction dataset.

When applying the instruction-tuning, the authors propose a relatively new data-collection framework that utilizes the knowledge-graph to make the instuctions reveal more causal knowledge and a data-sampling technique to overcome the data-inbalance of commonly- and uncommonly-researched proteins.

Experiments demonstrate the proposed methods can help LLMs achieve SOTA on several protein processing tasks.

**Strengths:**

- aligning protein 'language' with human language where LLMs originally trained on is one key and promising field to make LLMs solve protein processing tasks. In general, this paper can contribute to this field, i.e. achieving new SOTA by incremetally pre-training and collecting new data to conduct instruction finetuning.
- collecting instuction-tuning dataset from knowledge-graph is novel and insightful.
This can help to pose causal knowledge priori on finetuning LLMs.
The motivation is clearly clarified, and I think it can motivate further works in broader fields.
But the writing lacks details and citations of related works. I will detail the weakness in the next part.
- the data-sampling strategy is interesting and relatively new from my perspective. The experiments, althought with some weakness, demonstrate that the data-sampling strategy can heavily boost the performances.

**Weaknesses:**

- Regarding the data-collection. Prompt each triplet to ChatGPT and get the results as the instruction is an important step. But will this assume that ChatGPT can already be prompted to understand the protein input and make output correctly? Especially, in the Figure 4 which is an example of how ChatGPT is prompted, the ChatGPT can infer the information 'it is in the leptin family' and this information is important to pose the causal knowledge. How you guarantee that the output is correct and useful? 'ChatGPT' can understand protein is not a desiable assumption.

- Lacking of details. Regarding the data-collection, more details about the knowledge-graph need to be provided. You claim knowledge-graph can provide causal knowledge. The motivation can be more straghtforward if you tell what relationships the KG can provide and help the downstream tasks. Regarding the sampling, how the protein KG embedding is initialized? and what are the training details?

- More citations needed. The authors provide an overview of different instruction methods but I cannot see any related citations in the context. Also, regarding the sampling strategy, the authors should cite highly-related works in the context rather than putting all to the related-works far-away.

- More ablation studis needed. Ablation experiments on the incremental pre-training is necessary. And as  there is no enough details about training KG embeddings, I suspect the training is too tricky. The authors should provide ablations studies about only clusering by the editing distance to demonstrate the necessary of the embedding training. Also, ablation studies about the hyperparameters choices are needed.

**Questions:**

Please address my concerns list in the Weakness Part.

---

> ### Author Response · Authors · 2023-11-20
> **Rebuttal for Reviewer DyDT (1/2)**
>
> We would like to express our great appreciation to reviewer DyDT for their comments on our paper. The response to your concerns is as follow:
>
> > Regarding the data-collection. Prompt each triplet to ChatGPT and get the results as the instruction is an important step. But will this assume that ChatGPT can already be prompted to understand the protein input and make output correctly? Especially, in the Figure 4 which is an example of how ChatGPT is prompted, the ChatGPT can infer the information 'it is in the leptin family' and this information is important to pose the causal knowledge. How you guarantee that the output is correct and useful? 'ChatGPT' can understand protein is not a desiable assumption.
>
> The methods shown in Figure 3a and 3b assume that LLMs understand proteins and are able to come up with novel tasks and output desired answers. Compared to them, our innovation lies in the removal of such an assumption regarding LLMs possessing inherent protein-related knowledge, as depicted in Figure 3c. We retrieve the facts needed to generate the answer through KG and prompt LLMs how to generate tasks through KGC templates. All LLMs need to do is to faithfully convert the  triplet to natural language.
>
> In Figure 4, our retrieval process involves not only acquiring the triplet (MRCPG..., function, hormone) but also capturing associated knowledge (MRCPG..., family, leptin) that exhibits a causal relationship with the triplet. By utilizing these two interlinked pieces of information, we prompt ChatGPT to generate instructions. Figure 8 offers a detailed breakdown of the process involved in converting a triplet into instruction data. This breakdown encompasses inputs (prompt) and outlines ChatGPT's corresponding outputs.
> We will add these analysis in the final version.
>
> > Lacking of details. Regarding the data-collection, more details about the knowledge-graph need to be provided. You claim knowledge-graph can provide causal knowledge. The motivation can be more straghtforward if you tell what relationships the KG can provide and help the downstream tasks. Regarding the sampling, how the protein KG embedding is initialized? and what are the training details?
>
> Thanks for your advice. I would like to elaborate on the components involved:
>
> (1) **Knowledge Graph**: Our knowledge graph comprises triples denoting (protein, relation, annotation). As delineated in Section 4.1, the annotations encompass various attributes such as superfamily, family, domain, conserved site, active site, binding site, location, function, and involved biological processes of proteins. Notably, proteins utilized in downstream tasks have been excluded from the training data to maintain integrity.
>
> (2) **Causal Knowledge**: The mentioned annotations are distinct and independent entities. In the realm of protein function exploration by biologists, the customary approach involves commencing with easily studied features and deducing inferences based on the interconnections among these features. The objective of integrating causal knowledge is to equip language models with the capability to reason based on the relationships among annotations. This causal knowledge specifically encompasses the causal relationship from family and key sites to protein function and biological processes (A protein with an anaphylatoxin/fibulin domain indicates that it is probably located in an extracellular region).
>
> (3) **KG Embedding**: Following the TransE approach, we initiate embeddings for entities and relationships through a random initialization procedure [1]. We employ the SGD optimizer with a learning rate of 1.0. The dimensions of entities and relations' embeddings are set to 200. After 1000 epochs, the loss eventually converges to 0.168. The L2 distance utilized for clustering proteins is set to 1.4. We will add these details in the final draft.
>
> We will add these details in the final version.

---

> > ### Author Response · Authors · 2023-11-20
> > **Rebuttal for Reviewer DyDT (2/2)**
> >
> > > More citations needed. The authors provide an overview of different instruction methods but I cannot see any related citations in the context. Also, regarding the sampling strategy, the authors should cite highly-related works in the context rather than putting all to the related-works far-away.
> >
> > For seed task-based instruction methods, we have cited [2, 3, 6], and will cite [7] in final version. For raw document-based instruction method, we have cited [4] and would like additionally cite [5,8]. The success of seed task-based [2,3,6,7] and document-based [8] methods is based on the assumption that LLMs understand natural language. However, this assumption cannot be extended to protein languages, so [4,5] propose a template-based instruction generation method. Although this method avoids the problem of the model not understanding the protein, it will limit the diversity of the data. We propose the Knowledge Instruction framework, which ensures the diversity of data and avoids the problem of LLMs not understanding proteins through cooperation between LLMs and KGs.
> >
> > For sampling strategies, we will cite  [9, 10, 11]. Annotation imbalance naturally exists in datasets from diverse practical domains. Deep models are sensitive to this property and can be biased toward the dominant classes. Most of the previous works measure whether the data balance from the distribution of labels [9,10], ignoring the distribution of inputs. [11]  leverage the gradient norms’ magnitudes to identify the dominant classes, but this will introduce additional computational overhead. We take advantage of the characteristics of proteins to cluster simultaneously based on the inputs and annotations.
> > We will add these citations in the final version.
> >
> >
> > [1] Understanding the difﬁculty of training deep feedforward neural networks. AISTATS 2010
> > [2] Self-instruct: Aligning language models with self-generated instructions. 2023
> > [3] Unnatural instructions: Tuning language models with (almost) no human labor. 2023
> > [4] BioMedGPT: Open Multimodal Generative Pre-trained Transformer for Biomedicine. 2023
> > [5] Mol-Instructions: A Large-Scale Biomolecular Instruction Dataset for Large Language Models. 2023
> > [6] Generating datasets with pretrained language models. 2021
> > [7] Instruction induction: From few examples to natural language task descriptions. 2022
> > [8] Self-Alignment with Instruction Backtranslation. 2023
> > [9] Delving into Semantic Scale Imbalance. ICLR 2023.
> > [10] Graphsmote: Imbalanced node classification on graphs with graph neural networks. WSDM 2021.
> > [11] When Sparsity Meets Contrastive Models: Less Graph Data Can Bring Better Class-Balanced Representations. ICML 2023
> >
> > > More ablation studeis needed. Ablation experiments on the incremental pre-training is necessary. And as there is no enough details about training KG embeddings, I suspect the training is too tricky. The authors should provide ablations studies about only clusering by the editing distance to demonstrate the necessary of the embedding training. Also, ablation studies about the hyperparameters choices are needed.
> >
> > Thanks for the suggestion, we supplement the ablation experiments on InstructProtein(100m) as follows:
> >
> >  | KGE L2 Distance | Location (Sub) | GO (MF) |
> > |----- | ----- | ----- |
> > | 1.2 |  62.60 | 82.06 |
> > | 1.4 | 63.18 | 83.42 |
> > | 1.6 |  61.26 | 82.00 |
> > | 1.8 | 61.38 | 82.10|
> > | Edit Distance (5) | 62.22 |  81.74 |
> >
> > We will add these results in the final version.

---

### Official Review · Reviewer_modh · 2023-10-31

**Soundness:** 2 fair
**Presentation:** 3 good
**Contribution:** 3 good
**Rating:** 5
**Confidence:** 4

**Summary:**

The paper you described proposes InstructProtein, a large language model designed to bridge the gap between natural language and biological sequences, specifically proteins.

**Strengths:**

1. The paper introduces an approach to leverage large language models for comprehending biological sequences, such as proteins.

2. The paper introduces a knowledge graph-based instruction generation framework to construct a high-quality instruction dataset.

3. Experimental results show that InstructProtein outperforms state-of-the-art large language models.

**Weaknesses:**

1.It is mentioned in the text that the higher the absolute value of the pLDDT score, the better. However, I believe it is not that straightforward. The pLDDT score is an indicator of the quality of a protein's structure, and in the text, protein sequences are generated using a language model. The unstructured regions in the protein sequence have a significant impact on the protein's function, and correspondingly, the pLDDT score may be lower. To compare the quality of generated protein sequences, it's important to evaluate whether the distribution in multiple dimensions aligns with real proteins as shown in Figure 5. In addition, the visualized results should incorporate real protein visualization results.

2.In addition, There are alternative evaluation metrics for evaluating the generated sequences , such as assessments based on αβ structures, barrel helices, thermostability, metal element binding, etc. From an AI perspective, tools like Biopython can also be used to analyze biological aspects, such as pI values and electrostatic potential.

3.When conducting baseline comparisons, it is more convincing to compare against Mol-Instructions in addition to LLaMA and Alpaca. Mol-Instructions is a dataset constructed from text data of proteins based on UniProtKB, and fine-tuned on LLaMA-7B. Therefore, this paper should give the comparison results against Mol-Instructions.

4.In the section 4.2 , "For a fair comparison, we designed a template for each model through prompt engineering so that the model could follow our instructions and output the answers." What I understand is that for other models, they will fine-tune based on templates. Can you add supplementary information in the appendix regarding the hyperparameters and dataset size for model fine-tuning? This would make the results more reliable.

5.For ablation study, the article could delve into topics like how to address key residue mutations in overcoming the GO task or why the introduction of annotations leads to performance improvements. Additionally, the paper could elaborate on why fragment clusters based on protein properties can help avoid such issues.

**Questions:**

see above

---

> ### Author Response · Authors · 2023-11-20
> **Rebuttal for Reviewer modh (1/2)**
>
> We would like to express our great appreciation to reviewer modh for their comments on our paper. The unclear content have been revised in manuscript, and the response to your concerns is as follow:
>
> > It is mentioned in the text that the higher the absolute value of the pLDDT score, the better. However, I believe it is not that straightforward. The pLDDT score is an indicator of the quality of a protein's structure, and in the text, protein sequences are generated using a language model. The unstructured regions in the protein sequence have a significant impact on the protein's function, and correspondingly, the pLDDT score may be lower. To compare the quality of generated protein sequences, it's important to evaluate whether the distribution in multiple dimensions aligns with real proteins as shown in Figure 5. In addition, the visualized results should incorporate real protein visualization results.
>
> Regarding Figure 5, our evaluation focuses on the model's proficiency in adhering to structure-related instructions. In this context, the pLDDT score serves as a suitable evaluation metric. Specifically, when presented with an instruction such as 'I would like an all-alpha protein', the sequences generated by the model should demonstrate an increased proportion of alpha-helix and a decreased proportion of Intrinsically Disordered Protein (IDP) regions and other secondary structures. The high pLDDT score effectively reflects the low proportion of IDP regions, thus serving as a pertinent evaluation metric. Furthermore, Figures 5 (b,c) visually represent the embedding and predicted structures, highlighting significant differences in protein embeddings generated by distinct instructions. Notably, the folded structure aligns consistently with the provided instructions. In response to your suggestion, we've incorporated visualization results of real proteins, depicted in Figure 13. Upon observation, we note a striking similarity between the distribution of generated protein representations and that of real proteins. This congruence signifies the model's remarkable instruction-following ability. We will add these analysis in the final draft.
>
> > In addition, There are alternative evaluation metrics for evaluating the generated sequences, such as assessments based on αβ structures, barrel helices, thermostability, metal element binding, etc. From an AI perspective, tools like Biopython can also be used to analyze biological aspects, such as pI values and electrostatic potential.
>
> Thanks for providing alternative evaluation metrics for the generated sequences. The experiments is Section 4.3 are designed to demonstrate the model's instruction-following ability rather than merely assessing sequence quality. Therefore, the evaluation metrics need to correspond to the instructions (i.e., structure-related instruction with structure-related metrics, etc). Furthermore, in response to your suggestions, to assess generated sequences based on metal element binding, we additionally design three proteins with the instructions of ``I would like a protein that enables metal ion binding''. However, their binding ability cannot be concluded without wet experimental verification. So we compared it to existing protein to evaluate them as shown in Figure 15.
>
> > When conducting baseline comparisons, it is more convincing to compare against Mol-Instructions in addition to LLaMA and Alpaca. Mol-Instructions is a dataset constructed from text data of proteins based on UniProtKB, and fine-tuned on LLaMA-7B. Therefore, this paper should give the comparison results against Mol-Instructions.
>
> We leverage the checkpoint from "zjunlp/llama-molinst-protein-7b" and report the results as shown below:
>
> | Model (ACC)| Location (bin) | Location (Sub) | GO (BP) | GO (MF) | GO (CC) | Fold Rank (Fold) |
> | --- | --- | -- | --- | --- | --- | --- |
> | Mol-Instructions | 57.52 | 18.36   | 50.00 | 50.00 | 50.00 |  12.81 |
> | InstructProtein  | 85.19  | 70.79|71.49|85.83 |79.79 | 55.57 |
>
> From the results, we found three shortcomings of this model. Firstly, we have observed that employing a naive data cleaning strategy does not offer immunity to label imbalance issues. This observation further emphasizes the superiority and effectiveness of our proposed knowledge instruction methodology. Secondly, due to the lack of diversity of Mol-Instructions templates (excluding true/false questions), the model cannot understand the questions in the GO task and answer them accordingly. We propose that the cooperation of KG and LLMs to ensure the diversity of data is key for text and protein alignment. Finally, Mol-Instructions uses BPE to tokenize proteins, which makes it more difficult for the model to distinguish the nuances of proteins than amino acid-based tokenization, resulting in poor results on Fold Rank. Further, we leverage their model to design proteins as shown in Figure 14. The results demonstrate the model lacks the ability to follow structural instructions.

---

> > ### Author Response · Authors · 2023-11-20
> > **Rebuttal for Reviewer modh (2/2)**
> >
> > > In the section 4.2 , "For a fair comparison, we designed a template for each model through prompt engineering so that the model could follow our instructions and output the answers." What I understand is that for other models, they will fine-tune based on templates. Can you add supplementary information in the appendix regarding the hyperparameters and dataset size for model fine-tuning? This would make the results more reliable.
> >
> > Sorry for the misunderstanding, prompt engineering here refers to the process that an expert writes natural language instructions so that the model can output sentences that meet the requirements of the task. And we have presented the template in Appendix A.4. Note that designing templates is not trivial. For example, Galactica is trained on document-like corpus, and its prompt needs to conform to the format of the training document. The experiments we conducted are all in the zero-shot setting, meaning that we didn't fine-tune any parameters with any data.
> >
> > > For ablation study, the article could delve into topics like how to address key residue mutations in overcoming the GO task or why the introduction of annotations leads to performance improvements. Additionally, the paper could elaborate on why fragment clusters based on protein properties can help avoid such issues.
> >
> > We are sorry that we may not understand this concern. We did not invovle residue mutations in overcoming the GO task, nor to mention fragment clusters. So we are not sure how to delve into these questions by ablation study. As for the annotation introduction, does reviewer meaning the constructed instruction dataset? We ablate it and present the results are
> >
> > | model / ACC            | GO-BP | GO-MF | GO-CC |
> > | ---------------------- | ----- | ----- | ----- |
> > | Pre-train (UniRef100 + PubMed)                | 53.41 | 57.79 | 54.33 |
> > | Pre-train + Fine-tune (Knowledge Instruction)        | 71.49 | 85.83 | 79.69 |

---

### Official Review · Reviewer_Cqaz · 2023-11-02

**Soundness:** 2 fair
**Presentation:** 3 good
**Contribution:** 2 fair
**Rating:** 3
**Confidence:** 4

**Summary:**

This paper proposes a new model called InstructProtein that enables bidirectional generation capabilities between natural language and protein sequences within a single large language model (LLM). The key ideas are:
- Pre-train an LLM on both protein sequences and natural language text to acquire representations for both modes.
- Generate a  instruction dataset from a protein knowledge graph using a proposed debiased sampling strategy and knowledge causal modeling.
- Perform supervised instruction tuning on the LLM using the generated instructions to align the protein and natural language modalities.

The model is evaluated on classification tasks for protein sequence understanding tasks. Results show  improvements over open source LLMs.

**Strengths:**

- The idea of using instruction tuning to align proteins and text is innovative and enables bidirectional generation between the two modes. This opens up new possibilities for protein design and engineering.
- The proposed instruction generation methodology using knowledge graphs to improve instruction quality and the debiased sampling handles annotation imbalance and knowledge causal modeling provides causal links. These proposed method is able to improve instruction dataset quality.
- Comprehensive evaluations in classification tasks shows the improvement on the proposed methods.

**Weaknesses:**

-Additional baselines should be incorporated for a more comprehensive analysis. The current comparison lacks an evaluation against more advanced Large Language Models (LLMs) such as ChatGPT[1] , GPT-4 [2] , and Claude-2[3] . It is crucial to understand the performance disparities between the proposed method and these well-established models.

-The evaluation is restricted to classification tasks. There is a need to extend the assessment to more free-form instruction-based tasks[4] . One of the predominant applications of current chat assistant models is to interact with users. Hence, an evaluation that transcends beyond classification tasks is imperative to reflect more realistic usage scenarios.

-There is an absence of experimentation on the model's generalization capabilities. Recent studies in domain-specific instruction tuning suggest that training confined to a particular domain may impede generalization due to a lack of diversity in the training data [5]. I urge you to emphasize the aspect of 'diversity' in your study. Could you demonstrate the generalization abilities of your model in various contexts?

-Lack of scale up experiments. I am not sure if the conclusion holding in larger LLMs such as 7B,13B?

-Lack of experiments on different model family, such as LLaMA[6] and LLaMA-2[7].

[1] OpenAI. (2022). Introducing chatgpt. https://openai.com/blog/chatgpt, 2022. \
[2] OpenAI (2023). GPT-4 Technical Report \
[3] Anthropic (2022). Instroducing claude. https://www.anthropic.com/index/introducing-claude \
[4]  Dubois et al (2023). AlpacaFarm: A Simulation Framework for Methods that Learn from Human Feedback \
[5]  Zhang et al (2023). AlpaCare:Instruction-tuned Large Language Models for Medical Application \
[6]  Touvron et al (2023). LLaMA: Open and Efficient Foundation Language Models \
[7] Touvron et al (2023). Llama 2: Open Foundation and Fine-Tuned Chat Models

**Questions:**

- Can you show your results on compare with more powerful LLMs e.g ChatGPT, GPT-4, and Claude-2?

- Can you show results on some open-ending text generation tasks both in-domain and general domain. For general domain, I suggest to follow Alpaca-farm.

- Can you show results on different size of LLMs across different LLM families ?

---

> ### Author Response · Authors · 2023-11-20
> **Rebuttal for Reviewer Cqaz (1/2)**
>
> Thanks for your constructive and valuable comments, which have helped us improve the paper considerably. We will address these confusion and move this to the main text in the final version to make our paper stronger.
>
> Before the point-by-point response, we would like to underscore the contributions of our work. Firstly, our study identifies and highlights the prevalent issue of annotation imbalance within the current protein corpus, as elaborated in Section 2. We propose a comprehensive framework designed to generate high-quality text-protein-aligned instruction data sourced from an imbalanced knowledge base. We believe this instruction dataset will greatly faciliate future LLM research in biology. Secondly, we introduce InstructProtein, a model specifically trained on our constructed instruction data, demonstrating proficiency in comprehending the nuances of protein language.
>
> It is crucial to note that this article's primary focus centers on the translation between natural language and protein language. While our framework enables interaction with users through text-based communication, broader interaction akin to ChatGPT in general domain exceeds the current scope of our study.
>
> > Additional baselines should be incorporated for a more comprehensive analysis. The current comparison lacks an evaluation against more advanced Large Language Models (LLMs) such as ChatGPT , GPT-4, and Claude-2. It is crucial to understand the performance disparities between the proposed method and these well-established models.
>
> To comprehend the performance disparities between InstructProtein and these established models, we conducted the localization annotation experiments of 100 proteins. Presented below is the statistical results as well as a specific case along with the corresponding response:
>
> | model                      |ChatGPT |Claude-2| GPT-4 | InstructProtein |
> | ---------------------- | ----- | ----- | ----- | ---- |
> | Localization  (acc)   | 0.15 | 0.23 |  0.18   | 0.67 |
>
> ```
> User: What cellular component is the protein located in? Please give only one prediction. MKLAIGRLLSPLFLKNPQKPLIITKRFYSTPGERRIKDILTEKLSPSSLRVIDVSGGCGSMYQVAIKSKAFQGKNTLAQHRLVNSILKEEIRNMHGLNLSTEVEDDISAGGSTTSS
>
> ChatGPT: Based on the provided amino acid sequence, the prediction for the cellular component of the protein is the cell membrane. (Wrong)
>
> GPT4: Based on the sequence provided, I would predict that this protein is located in the mitochondrion.  Here's the rationale for my prediction: ...(Wrong)
>
> Claude-2: Nucleus. My rationale: (1) It lacks a signal peptide, indicating it is not a secreted protein. (2) It does not contain any predicted transmembrane domains, suggesting a soluble localization. (3) It has abundant lysine and arginine residues providing a positive charge. (Wrong)
>
> InstructProtein: The protein is located in mitochondrion. (Correct)
> ```
>
> The results yield three pivotal observations. Firstly, we are surprised to discover that ChatGPT and Claude-2's predictions solely encompassed cytoplasm, nucleus, cell membrane, and secreted locations. This finding underscores that annotation imbalance impacts even these well-established models. Notably, the performance degradation due to data imbalance persists despite model enhancements or increased training data, further emphasizing the significance of our contributions of the high-quality instruction dataset. Secondly, while the responses from them may initially appear plausible, their prediction accuracy is notably deficient. This highlights a crucial aspect regarding the precision of their predictions, indicating a large performance disparity. Additionally, we employed ChatGPT and Claude-2 to design the globin-like, immunoglobulin-like, and tim beta-alpha-barrel protein, and the outcomes are visually presented in Figure 14. We observed that Claude-2 ignores all instructions and output proteins with similar structures no matter what instructions were given. ChatGPT, on the other hand, can only understand basic instructions (globin-like, immunoglobulin-like). Only InstructProtein can faithfully follow instructions to generate desired proteins.
>
> Here, we would like to explain why we did not consider ChatGPT or GPT-4 as baselines before. Until the day we submitted this manuscript to ICLR, ChatGPT often refused to generate protein sequences, as shown in Figure 1. We have tried our best to write the appropriate prompt, but for some cases, these LLMs still tend to decline protein sequence generation, which makes automated benchmark comparison impossible.

---

> > ### Author Response · Authors · 2023-11-20
> > **Rebuttal for Reviewer Cqaz (2/2)**
> >
> > > The evaluation is restricted to classification tasks. There is a need to extend the 	assessment to more free-form instruction-based tasks. One of the predominant applications of current chat assistant models is to interact with users. Hence, an evaluation that transcends beyond classification tasks is imperative to reflect more realistic usage scenarios.
> >
> > InstructProtein stands as a model facilitating bidirectional alignment between natural language and protein language. We categorize protein-related scenarios into two domains: describing protein properties and generating proteins with specified properties. Evaluating the accuracy of the model's predictions through classification tasks stands as the most effective approach to gauge the quality of descriptions. This evaluation methodology aligns with prevalent practices in the field of Natural Language Processing (NLP), evident in benchmarks like GLUE, superGLUE, and the Big-bench benchmark.
> >
> > More importantly, it's essential to note that our evaluation extends beyond mere classification tasks. In Section 4.3, the evaluation encompasses its capacity for designing protein sequences. It's noteworthy that, to our knowledge, InstructProtein is the pioneering model capable of designing proteins based on natural language instructions.
> >
> > > There is an absence of experimentation on the model's generalization capabilities. Recent studies in domain-specific instruction tuning suggest that training confined to a particular domain may impede generalization due to a lack of diversity in the training data. I urge you to emphasize the aspect of 'diversity' in your study. Could you demonstrate the generalization abilities of your model in various contexts?
> >
> > In the realm of protein science, the generalization capability can be examined from two aspects: (1) The model's proficiency across a diverse array of protein language-related tasks. In Table 2, we showcase the model's competence in describing protein functions, while Figures 5 and 6 illustrate its capability to generate proteins with various required properties. (2) The model's accuracy in predicting out-of-domain functions. Taking the catalytic reaction classifcation as an example, notably, our training dataset lacks catalytic reaction annotations for proteins. We present the results below to demonstrate the model's performance in this task:
> >
> > | model         |OPT |Galactica|InstructProtein |
> > | ------------- | ----- | ----- | ----- |
> > | Reaction  (F1) |  60.02  |  61.24 | 74.04 |
> >
> > These results indicate the generalization abilities of our model, and we will add them in the final version. We also posed these questions to ChatGPT, and it turned out that ChatGPT was unable to determine the relationship between a protein and the reaction it could catalyze. Its answers are all negative as shown in Figure 16.
> >
> > > Lack of scale up experiments. I am not sure if the conclusion holding in larger LLMs such as 7B,13B? Lack of experiments on different model family, such as LLaMA[6] and LLaMA-2[7].
> >
> > Due to limited rebuttal time, we can only conduct experiments on OPT-6.7B and LLaMA 7B, and the results are shown below:
> >
> > | Model | GO-BP | GO-MF | GO-CC |
> > | ----- | ----- | ----- | ----- |
> > | Alpaca (LLaMA-7B) | 61.69 | 59.37 | 57.98 |
> > | InstructProtein (OPT-1.3B) | 71.49 | 85.83 | 79.79 |
> > | InstructProtein (OPT-6.7B) | 73.94 | 87.95 | 82.37 |
> > | InstructProtein (LLaMA-7B) | 75.52 |  87.44| 83.41 |
> >
> > We can observe that as the size of the model increases, the better the model's understanding of the protein sequence. We will add more scalability experiments in the final version of this paper.

---

### Official Review · Reviewer_Ex2Y · 2023-11-02

**Soundness:** 2 fair
**Presentation:** 3 good
**Contribution:** 3 good
**Rating:** 6
**Confidence:** 4

**Summary:**

In this work, the authors developed LLM called InstructProtein that has bidirectional generation capabilities: (1) translating the protein sequence to its textual function description (2) translating natural language instruction to the protein sequence.
To achieve this, the LLM is pre-trained on protein and natural language corpora. Specifically, the protein UniRef100 and sentences from PubMed abstracts are used for pre-training. To further obtain good performance on downstream tasks, the authors constructed a high-quality instruction dataset. This dataset is generated based on the knowledge graph that is constructed from the annotations of UniProt/Swiss-Prot. Overall, a KG triple to instruction generator (based on ChatGPT) was used . The chain-of-thought strategy and debiased sampling strategy were used for this generation process. In total, 2.8 million data is constructed.
Experiments on de novo design and three protein function classification tasks (1) Protein localization prediction (2) Protein function annotation (3) Metal Ion Binding prediction showed that the protein knowledge instructions can boost the performance of protein understanding and design tasks.

**Strengths:**

This is an interesting work on applying large language model to bioinformatics. Some related work have tried to align protein with human language. However, they either only showed unidirectional cross-modal capability, focusing solely on converting protein language to texts, or did not align the protein and human language very well. InstructProtein improves this by pre-training on UniRef100 and sentences from PubMed abstracts, providing a good foundation model on protein domain and filling the gap between the two languages and enabling the bidirectional generation. This work also contributes the first high-quality protein instruction dataset, by designing an effective data generation framework.

**Weaknesses:**

For the first conclusion in 4.2: The results (comparing with OPT, LlaMa, Alpaca) demonstrate that training with the corpus where proteins and natural language coexist is beneficial to LLMs, enhancing their proficiency in protein language understanding. However, I think this argument can not be concluded based on these results. Because the performance of the InstructProtein is contributed by both pre-training and finetuning. Without finetuning LLMs on the instruction corpus, we can not conclude that the coexist of proteins and natural language is beneficial (even this conclusion is quite intuitive).

**Questions:**

Do we have any quality evaluation on the generated instruction dataset?

---

> ### Author Response · Authors · 2023-11-20
> **Rebuttal for Reviewer Ex2Y**
>
> We thank Reviewer Ex2Y for reviewing our paper and providing thoughtful feedback on our work. We have revised the manuscript to make this paper easier to follow. Here, we provide details on comments below.
>
> > For the first conclusion in 4.2: The results (comparing with OPT, LlaMa, Alpaca) demonstrate that training with the corpus where proteins and natural language coexist is beneficial to LLMs, enhancing their proficiency in protein language understanding. However, I think this argument can not be concluded based on these results. Because the performance of the InstructProtein is contributed by both pre-training and finetuning. Without finetuning LLMs on the instruction corpus, we can not conclude that the coexist of proteins and natural language is beneficial (even this conclusion is quite intuitive).
>
> Thank you for highlighting the potential misunderstandings that the conclusion might induce. The essence of this conclusion lies in highlighting the limitation of the current Natural Language Processing (NLP) corpus in furnishing protein comprehension abilities to Large Language Models (LLMs). To further substantiate the advantages of a corpus integrating both proteins and natural languages, we are providing additional experimental results, presented below:
>
> | model / ACC            | GO-BP | GO-MF | GO-CC |
> | ---------------------- | ----- | ----- | ----- |
> | Pre-train (UniRef100 + PubMed)                | 53.41 | 57.79 | 54.33 |
> | Pre-train + Fine-tune (Knowledge Instruction)        | 71.49 | 85.83 | 79.69 |
>
> The pre-training is conducted with the UniRef100 and Pubmed datasets that respectively contain proteins  and biomedical literature, while the finetuning is conducted using the proposed Knowledge Instruction approach with aligned natural language and protein language corpora. We can observe that the improvement of the performance is mainly due to the instruction tuning stage. We will add this results to the final draft.
>
> > Do we have any quality evaluation on the generated instruction dataset?
>
> We advocate assessing the quality of the generated instruction dataset based on both correctness and diversity. To mitigate the potential generation of erroneous data by Large Language Models (LLMs), we construct a Knowledge Graph (KG) based on the well-maintained UniProbKB database and regared it as a reliable knowledge source. Then, we sample from this KG and convert the sampled triples to instructions. In this way, the correstness of the instruction dataset can be tracked and guaranteed by the UniProbKB.
> Diversity is quantified with information entropy. For instance, considering subcellular location-related instructions, Figure 2 highlights annotation imbalance. The original data distribution's information entropy is calculated as $H(x)=-\sum_x p(x)\log p(x) = 3.17$. Through our proposed KG triplet retrieval method, this entropy increases to $3.68$. This increment in information entropy denotes enhanced diversity within the dataset, illustrating increased coverage and richness of information in the generated instructions. We will add the above analyses to the final draft.

---

### Author Response · Authors · 2023-11-23
**Urgent Request for Re-review and Discussion**

Dear Reviewers and AC,

We genuinely value the constructive comments and insightful suggestions you provided for our work. Recognizing the approaching end of the discussion period on November 22nd, we kindly urge you to participate in the ongoing discussion and provide any additional insights or clarifications you may have. Your expertise is invaluable to us, and we believe your input will significantly contribute to the improvement of our work.

Thank you very much for your time and consideration. We look forward to hearing from you soon.

Authors